# Integrated Transcriptomic and Proteomic Analyses Reveal Molecular Mechanism of Response to Heat Shock in *Morchella sextelata*

**DOI:** 10.3390/jof11010076

**Published:** 2025-01-18

**Authors:** Jiexiong Zhang, Yanxia Li, Yifan Mao, Yesheng Zhang, Botong Zhou, Wei Liu, Wen Wang, Chen Zhang

**Affiliations:** 1School of Ecology and Environment, Northwestern Polytechnical University, Xi’an 710072, China; zhangjx_light@mail.nwpu.edu.cn (J.Z.); myf_nwpu@163.com (Y.M.); zhoubotong@mail.nwpu.edu.cn (B.Z.); 2Shandong Junsheng Biotechnologies Co., Ltd., Liaocheng 252400, China; dongyue899205@163.com (Y.L.); zhangyesheng1987@163.com (Y.Z.); 3The Germplasm Bank of Wild Species, Yunnan Key Laboratory for Fungal Diversity and Green Development, Kunming Institute of Botany, Chinese Academy of Sciences, Kunming 650201, China; zhenpingliuwei@163.com

**Keywords:** thermotolerance, *Morchella sextelata*, CWI, HSPs, antioxidant, ribosome, transcriptome, proteome

## Abstract

Morels (*Morchella* spp.), as one of the rare macroascomycetes that can be cultivated artificially, possess significant economic and scientific values. Morel cultivation is highly sensitive to elevated temperatures; however, the mechanisms of their response to heat shock remain poorly understood. This study integrated transcriptomic and quantitative proteomic analyses of two *M. sextelata* strains with different thermotolerance (labeled as strains C and D) under normal (18 °C) and high temperature (28 °C) conditions. From over 9300 transcripts and 5000 proteins, both consistency and heterogeneity were found in response to heat shock between the two strains. Both strains displayed a capacity to maintain cellular homeostasis in response to heat shock through highly expressed cell wall integrity (CWI) pathways, heat shock proteins (HSPs), and antioxidant systems. However, strain D, which exhibited stronger thermotolerance, specifically upregulated the ubiquitin ligase *Rsp5*, thereby further promoting the expression of HSPs, which may be a key factor influencing the thermotolerance difference among *M. sextelata* strains. A conceptual model of the heat shock adaptation regulatory network in *M. sextelata* was proposed for the first time; the results provide novel insights into the thermotolerance response mechanisms of macroascomycetes and valuable resources for the breeding enhancement of thermotolerant morel strains.

## 1. Introduction

Morels are one of the rare macroascomycetes that can be artificially cultivated and are globally recognized for their excellent edible and medicinal properties [1,2]. In recent years, cultivation technology characterized by exogenous nutrient bags has made significant progress in China, particularly for saprotrophic species such as *M. sextelata*, *M. exima*, and *M. importuna* [3,4,5]. These techniques have gradually matured and demonstrate great market potential [1,5,6]. These cultivable species are either obligate pyrophytic (*M. sextelata* and *M. exima*) or facultatively pyrophytic species (*M. importuna*), and all of them can fruit in post-fire habitats [7,8]. Despite this adaptation, morel cultivation is particularly susceptible to high temperatures, requiring an ideal cultivation temperature below 20 °C [5]. Due to global warming, not only has the global temperature increase exceeded 1.5 °C, but the frequency of extremely hot days has also significantly increased compared to the past [9,10]. Recently, there have been consistent reports about severe losses in morel cultivation caused by high temperatures [6]. Against the backdrop of global warming, high temperatures are increasingly becoming a significant impediment to the sustainable development of the artificial cultivation of *Morchella*.

High-temperature represents a major abiotic stress factor that can significantly affect the growth of fungi, disrupt cellular homeostasis, trigger apoptosis, and alter the synthesis of secondary metabolites [11,12]. The impact of heat shock on the cultivation of edible mushrooms is becoming increasingly serious [13,14]. The current research on the molecular mechanisms of edible mushrooms adapting to heat shock mainly focused on the basidiomycetes and has revealed a range of response pathways. Studies have shown that para-aminobenzoic acid significantly enhances the thermotolerance of *Agaricus bisporus* by regulating heat shock proteins and antioxidant systems [15]. In *Ganoderma lucidum*, heat shock regulates mycelial growth through a soluble Ca^2+^-mediated signaling pathway [16]. In *Pleurotus ostreatus*, trehalose induced by reactive oxygen species (ROS) under heat shock is crucial for its survival [17]. However, studies on the adaptation mechanisms of macroascomycetes to heat shock remain scarce, with only a few reports, such as the impact of heat shock on the metabolic products of *Cordyceps militaris* [18]. In morel, preliminary research has found that heat shock induces significant physiological and morphological changes, including increased hyphal branching, reduced biomass accumulation, decreased growth rate, and premature aging [19]. At the molecular level, heat shock causes the accumulation of reactive oxygen species, enhanced activity of antioxidant enzymes, and changes in tyrosine metabolism [19]. Despite these previous studies, the understanding of the systematic molecular regulatory network of the morel thermotolerance response remains fragmented and insufficient. Consequently, there is an urgent need for comprehensive research in this area.

To address this issue, two *M. sextelata* strains (labeled as strains C and D) with different thermotolerance were selected, which were verified through laboratory mycelial heat shock screening and field cultivation validation. Then, a mycelium subculture of the two strains was conducted under high (H, 28 °C) and normal (N, 18 °C) temperature conditions. By integrated transcriptomic and proteomic analyses, we conducted a comprehensive investigation into the molecular regulatory mechanisms and networks of heat shock on *M. sextelata* mycelium. The results will not only advance our understanding of fungal stress biology but also provide valuable insights for improving the thermotolerance of *M. sextelata.*

## 2. Materials and Methods

### 2.1. Sample Collection and Tissue Isolation

Strains C and D were isolated from fruiting bodies collected from natural habitats. Strain C was collected from Longyang District, Baoshan City, Yunnan Province (coordinates: 99.32° E, 24.87° N; elevation: 1888 m a.s.l.), while strain D was obtained from Bomi County, Nyingchi City, Tibet Autonomous Region (coordinates: 94.91° E, 30.18° N; elevation: 2217 m a.s.l.). On the day of collection, the fruiting bodies were immediately bisected longitudinally along their vertical axis and dehydrated using a cold-air ventilation dryer, followed by storage in a cool, dark environment.

The strains were isolated from fruiting bodies through tissue isolation. For the isolation procedure, a small tissue block (approximately 0.5 × 0.5 cm^2^) was excised from the central portion of the collected fruiting body stipe using a sterile scalpel and immersed in sterile water with gentle agitation to remove surface contaminants. Surface sterilization was then performed by immersing the cleaned tissue in 75% ethanol for 1.0–1.5 min. The sterilized tissue was subsequently sectioned into small fragments (0.2 × 0.3 cm^2^) and transferred onto Potato Dextrose Agar (PDA) medium (composed of 200 g boiled potato extract, 20 g glucose, 20 g agar powder, and 1 L distilled water). The cultures were incubated in darkness at 23 °C. Once the colonies reached approximately 1.5 cm in diameter, hyphal tips were subcultured onto fresh PDA medium.

### 2.2. Temperature Selection and Heat Shock Treatment

Previous studies have shown that the optimal temperature range for morel mycelial growth is around 20 °C, although there is no consensus on the exact optimal cultivation temperature [20], possibly due to variations in temperature adaptability among different morel strains. In field cultivation, temperatures below 20 °C are required for morel growth; thus, 18 °C was selected as the optimal growth temperature for this study. Both the growth rate and biomass of morel mycelia begin to decline at temperatures between 25 and 30 °C [21], indicating significant thermal stress on mycelia growth at these temperatures. Since temperatures between 25 and 30 °C are commonly encountered in field cultivation, we tested strains C and D at one-degree intervals from 26 °C to 30 °C (Appendix A). Both strains maintained continuous growth for over 10 generations at 26 °C and 27 °C. However, starting from 28 °C, the number of sustainable generations decreased dramatically. Strain C could sustain growth for six, four, and three generations at 28 °C, 29 °C, and 30 °C, respectively, while strain D sustained for eight, six, and three generations. These observations indicate that both strains began experiencing significantly increased thermal stress at 28 °C; therefore, we selected 28 °C as the heat shock temperature for this study.

This study used 18 °C as the control temperature and 28 °C as the heat shock condition. Strains C and D were inoculated at the edges of PDA culture medium and placed in an incubator set to the corresponding temperatures, with five biological replicates for each strain. When the mycelium in a culture medium grows close to the opposite end, or when a single generation (abbreviated to G) of cultivation exceeds 6 days, subculturing is performed. During each subculturing, agar blocks are taken from the leading end of the mycelium in the current culture medium and inoculated onto the edge of a next medium to continue cultivation until the mycelium stops growing. Additionally, sterile glass papers were placed on the culture medium to facilitate the collection of mycelia for sequencing. After subculturing, all mycelial samples are collected, uniformly placed into cryovials, and immediately frozen in liquid nitrogen, followed by storage at −80 °C. The mycelium from the 1st generation (G1, representing the early stage), 3rd generation (G3, representing the middle stage), and 5th generation (G5, representing the late stage) are subjected to transcriptomic and proteomic analyses. In subsequent analyses, all samples were categorized into six groups: N1, N2, and N3 for normal temperature conditions, and H1, H2, and H3 for high temperature conditions, with the numbers 1, 2, and 3 representing the early, middle, and late stages of culture, respectively.

### 2.3. Field Cultivation Validation

Based on the cultivation method summarized by Liu et al. (2017) [3], we further standardized key parameters, including air humidity, temperature, and soil type, while providing detailed management protocols (Appendix A). The cultivation experiments were conducted at two locations: Baoshan City, Yunnan Province (coordinates: 99.11° E, 25.04° N; elevation: 1982 m a.s.l.; temperature range: −0.7 °C to 26.1 °C) and Liaocheng City, Shandong Province (coordinates: 115.50° E, 36.55° N; elevation: 41 m a.s.l.; temperature range: 0.6 °C to 18.9 °C). Each location had a cultivation area of 0.12 hectares.

Although the soil types differed between the two sites—paddy soil derived from red soil in Baoshan and sandy loam in Liaocheng—both soils supported normal agricultural activities, ensuring basic physicochemical properties suitable for cultivation. Due to regional climatic differences, distinct cultivation systems were implemented at each site. Simple shade-net greenhouses were used in Baoshan, China, while temperature-controlled greenhouses were employed in Liaocheng.

Temperature data throughout the cultivation period were collected using ground-level temperature sensors for indoor temperatures and local meteorological stations for outdoor conditions. While outdoor temperatures in Liaocheng ranged from −11 °C to 22 °C, the greenhouse maintained temperatures between 3.5 °C and 18.9 °C, with a daily temperature fluctuation of 7.4 °C, demonstrating effective temperature regulation and cold protection. In contrast, the simple shade-net greenhouses in Baoshan provided limited temperature control, with temperatures ranging from −0.7 °C to 26.1 °C and a significantly higher daily temperature fluctuation of 12.2 °C compared to Liaocheng. The detailed temperature data are supplied in Appendix A.

Yield data were obtained by measuring the total fresh weight of all fruiting bodies produced throughout the cultivation cycle, calculated as the average yield per square meter.

### 2.4. Genome Annotation

The reference genome used in this study was downloaded from *M. sextelata* assembly ASM2471366v1 (https://www.ncbi.nlm.nih.gov/datasets/genome/GCA_024713665.1/, accessed on 1 May 2023) [22]. Due to the lack of gene structure annotation in the submission, we employed de novo strategies for gene annotation by Augustus (version 3.3.3) [23]. Gene function annotation was performed through the Gene Ontology (GO) database and the Kyoto Encyclopedia of Genes and Genomes (KEGG) database, employing BLAST (version 5.1) [24].

### 2.5. RNA Extraction, Library Construction, Sequencing, and Data Processing

Total RNA was extracted from collected samples using the CTAB method [25]. The concentration of the extracted nucleic acids was measured using the Nanodrop2000 (manufacturer: Thermo Fisher, Waltham, MA, USA; model: Nanodrop2000), and the integrity was assessed using the Agilent 2100, Santa Clara, CA, USA, LabChip GX (manufacturer: PerkinElmer; model: PerkinElmer LabChip GX). Library preparation was conducted using the TruSeq™ RNA Sample Preparation Kit (Illumina, San Diego, CA, USA), and quality control was performed using the Qsep-400 method. Subsequently, the PE150 strategy was carried out on the Illumina NovaSeq 6000 platform (Illumina, San Diego, CA, USA), and approximately 5.86 Gb of raw data were arranged.

The raw data were processed to remove adapters and low-quality reads using fastp (version 0.21.0) [26], and qualified reads were mapped to the reference genome (ASM2471366v1) using Hisat2 (version 2.1.0) [27]. Reads for each transcript were quantified to obtain read counts using htseq2 (version 2.2.1) [28], and fragments per kilobase of exon model per million mapped fragments (FPKM) values were obtained using a Python (version 3.8.5) script for subsequent comparative analysis.

### 2.6. Proteome Extraction, LC-MS/MS Analysis, Database Searching, and Data Processing

Mycelial samples were first mechanically disrupted using a high-throughput tissue grinder (HF-48, Shanghai Hefan Instruments, Shanghai, China) with steel beads. The samples were then further homogenized by ultrasonic disruption (JY96-IIN, Shanghai Huxi Industrial, Shanghai, China) on ice using the following parameters: 300 W, 3 s on/3 s off pulse cycles for 5 min. The homogenized samples were lysed in a buffer containing 8 M urea, 30 mM 4-(2-hydroxyethyl)-1-piperazineethanesulfonic acid (HEPES), 1 mM phenylmethylsulfonyl fluoride (PMSF), 2 mM ethylenediaminetetraacetic acid (EDTA), and 10 mM dithiothreitol (DTT). After centrifugation at 20,000× *g* for 30 min, the supernatant was treated with 10 mM DTT for reduction and 55 mM iodoacetamide for alkylation, followed by enzymatic digestion with trypsin and desalting using a C18 column. Peptide quantification was performed using the bicinchoninic acid (BCA; Pierce Quantitative Colorimetric Peptide Assay) or Qubit (Thermo Scientific) system.

LC-MS/MS analysis was conducted on a timsTOF MS (Bruker, Billerica, MA, USA) equipped with a high-performance liquid chromatography system, coupled with a nanoElute^®^ (Bruker, Billerica, MA, USA). Peptides were loaded onto a homemade column (75 μm × 250 mm; 1.9 μm ReproSil-Pur C18 beads, Dr. Maisch GmbH, Ammerbuch), which was heated to 60 °C and separated using a 60-min gradient from 2% to 80% mobile phase B at a flow rate of 300 nL/min. Mobile phases A and B were 0.1% (*v*/*v*) formic acid/ddH_2_O and 0.1% (*v*/*v*) formic acid/acetonitrile, respectively. The mass spectrometer was operated in the Data Independent Acquisition (DIA) Parallel Accumulation-Serial Fragmentation (PASEF) model [29]. Fragmentation analysis was divided into 64 × 26 Th precursor isolation windows, ranging from *m*/*z* 400 to 1200, with a 1 Th isolation window overlap. Collision energy varied linearly from 59 eV at 1/K0 = 1.6 Vs cm^−2^ to 20 eV at 1/K0 = 0.6 Vs cm^−2^ based on mobility. A spectral library was constructed using data-dependent acquisition (DDA) data and DIA data, and raw quantitative proteomic data were obtained using the Pulsar algorithm embedded in Spectronaut (Biognosys, Schlieren, Switzerland) for data searching, followed by normalization using the upper quartile.

We retained proteins detected in at least two out of three replicates for data imputation. Data imputation was initially performed using the tool NAguideR [30] to handle missing data with 23 different analytical methods. Four classic evaluation criteria—normalized root mean square error (NRMSE), “sum of ranks based on NRMSE (SOR)”, “sum of prediction errors squared (PSS)”, and “average correlation coefficient between original and calculated values (ACC_OI)”—were used to evaluate and rank these 23 methods (evaluation results are provided in the Appendix A). Finally, ImputeSeqRob [31] model ranked first, and its results were selected for the imputation of missing values. The imputed expression matrix was then used for subsequent comparative analysis.

### 2.7. Respective Comparative Analysis in Transcriptomics and Proteomics

In the transcriptome analysis, DESeq2 [32] was utilized for differential expression analysis, with the threshold criteria for selecting differentially expressed genes (DEGs) being |log_2_(Fold Change(FC))| ≥ 1, *p* value ≤ 0.05. Proteomic differential expression analysis was performed using by *t*-test in Python, with the threshold criteria for selecting differentially expressed proteins (DEPs) being FC ≥ 1.2 or FC ≤ 0.83, *p* value ≤ 0.05. Clustering analysis of genes or proteins to explore the time-series expression patterns was conducted by R package Mfuzz [33], which is based on soft clustering analysis. The input files for Mfuzz clustering analysis were the expression matrices of genes or proteins, derived from genes or proteins that exhibited differential expression in at least one period. These genes or proteins were analyzed for their time-series expression patterns during periods H1, H2, and H3 using Mfuzz, with the mean expression levels of the samples from period N1 as the reference. Subsequent functional pathway analysis visualization was completed using R (version 4.3.1) scripts based on GO and KEGG annotations.

### 2.8. Integrated Comparative Analysis of Transcriptomics and Proteomics

The correlation between the transcriptome and proteome of each sample is calculated using the Spearman’s rank correlation test. Then, the nine-quadrant plots were completed using R tools, with the threshold criteria at |log_2_FC| ≥ 1 for the transcriptome and |log_2_FC| ≥ 0.263 for the proteome. The heatmap data for specific gene modules were derived from the log_2_FC values obtained by comparing the FPKM and protein expression levels of the genes between H and N conditions. Orthologous genes were identified by OrthoFinder (version 2.5.4) [24], and the table of corresponding orthologous genes can be found in Appendix A. The differential expression analysis of genes and proteins related to the cell wall integrity (CWI) pathway and heat shock proteins (HSPs) under heat shock was performed using DESeq2 (version 1.30.1) [32], with *p* value ≤ 0.05. Subsequently, the enrichment of CWI-related and HSP-related genes within the differentially expressed gene set was assessed using Fisher’s exact test. Subsequently, the assessment of significant enrichment of CWI genes and HSP genes/proteins was completed using Fisher’s test, with the background being the set of genes/proteins that are significantly upregulated under high-temperature conditions with *p* value ≤ 0.05.

### 2.9. Comparative Analysis of Different Thermotolerance Between M. sextelata Strains

In this study, the log_2_FC of the transcription/protein expression levels between strains D and C under identical conditions (cultivation stage, temperature) was used to determine the relative expression levels (RELs) between the two strains. The Mfuzz [33] clustering algorithm, using the REL of genes and proteins between the two strains during period N1 as the initial values, analyzed the temporal trends of the REL values for the two strains across three periods under high-temperature conditions, removing features with a standard deviation less than 0.5. The number of clusters was set to 24, and clusters with upregulated expression trends were manually selected for subsequent functional enrichment analysis. The method of functional enrichment analysis was same as that mentioned for the transcriptome and was implemented through R scripts for visualization.

Weighted Gene Correlation Network Analysis (WGCNA) [34] was performed using the WGCNA (version 1.72-5) R package. Co-expression networks were constructed using normalized transcriptomic/proteomic data. The network type was specified as signed hybrid, and the best soft-thresholding power (β) was determined by examining the scale-free topology of the network and the average connectivity within different β value ranges. The minimum module size was set to 30, and the Pearson correlation cut-off height was ≥0.25. All proteins were referenced against *Saccharomyces cerevisiae*, and protein network graphs were constructed using the STRING database (version 11.0) [35]. The top 30 proteins were extracted based on the EcCentricity algorithm using Cytoscape (version 3.9.195) [36], and the protein network graphs were redrawn and subjected to DBSCAN [37] clustering analysis according to the STRING database [35].

To validate the expression levels of key genes, frozen stocks of strain D and strain C were retrieved from −80 °C storage and re-cultured on PDA medium (covered with cellophane) for 6 days under the corresponding temperature conditions. Mycelia were collected and RNA was extracted using the HiPure Fungal RNA Kit (Magen, Guangzhou, China). The real-time polymerase chain reaction (RT-PCR) experiments were conducted using the Roche480 II real-time PCR system (Roche, Basel, Switzerland) according to the manufacturer’s instructions, with the *Cyc3* gene as an internal reference [38]. The primers used are shown in Appendix A.

### 2.10. Western Blot

Inoculate strain C onto petri dishes lined with sterile glass papers, and then cultivate continuously at 18 °C (normal temperature) and 28 °C (heat shock temperature) for 6 days. Scrape the cultivated mycelium from the dishes, and quickly freeze them in liquid nitrogen. After freezing, grind the samples into a powder, and add RIPA (Radio-Immuno Precipitation Assay) buffer to lyse the cells for 1 h followed by centrifugation. After determining the protein concentration, normalize the protein amounts, and place the proteins in a constant temperature device at 100 °C for 10 min. Add the protein maker and protein samples to the SDS-PAGE gel for electrophoresis separation, and transfer to a PVDF (Polyvinylidene Difluoride Flask) membrane. Block the membrane with milk for 30 min, then incubate overnight with the Hsp60 antibody. Wash with PBST (Phosphate-Buffered Saline with Tween 20) three times, each for 10 min, and then incubate the membrane with the corresponding secondary antibody at 37 °C for 1 h before discarding the secondary antibody solution. Subsequently, wash with TBST (Tris(hydroxymethyl)Amine-Buffered Saline) three times, each for 10 min. Finally, after draining the moisture, apply ECL luminescence solution evenly, and place the membrane in a chemiluminescent imaging device for imaging.

## 3. Results

### 3.1. Morphological Phenotypes and Yield Variations of Morel Mycelium Under Heat Shock

The mycelium of strains C and D was cultured at 18 °C and 28 °C, respectively. At the normal temperature (18 °C), there was no significant difference between the two strains. The mycelium of both strains was pale white, growing radially from the inoculation block on the plate with a neat edge (Figure 1a), with a growth rate of 0.4–0.5 mm/h (Figure 1c).

Under high-temperature stress at 28 °C, the growth rate of both strains decreased markedly, with the fastest growth rate only 0.3 ± 0.02 mm/h (Appendix A). There was also a difference in the number of generations that could maintain growth under high temperature for the two strains, with six generations for strain C and nine generations for strain D (Figure 1a,c), suggesting that strain D possessed a greater capacity for temperature tolerance. In phenotypes, both strains showed some consistency under heat shock. The *M. sextelata* mycelium exhibited as slightly yellowish at 28 °C, and the radial growth of mycelium along the inoculation block was not obvious. The edge smoothness of the mycelial growth was poor, and, in the later stages of heat shock, the edge of the mycelium obviously turned dark brown (Figure 1a). Microscopic observations have found that, under continuous high-temperature stress, the process of branching changes in the mycelium is similar to the aging of morel mycelium [33]. At normal temperatures, there is a higher density of mycelium branches with smaller branch angles, while, under high temperatures, the number of branches decreases and, after continuous heat shock, the branch angles of strain enlarged, shifting from an acute angle to nearly 90 degrees (Figure 1b).

Further verification of the strains’ thermotolerance was conducted through field cultivation trials. In the 2021–2022 and 2022–2023 cultivation seasons, cultivation trials for these two strains were conducted in two distinct environments: Baoshan City, Yunnan Province, which experienced high-temperature stress with temperatures reaching up to 26.1 °C, and the suitable temperature conditions of the greenhouses in Liaocheng City, Shandong Province, where temperatures were capped at 18.9 °C. The yield data statistics from both locations showed that, in the suitable temperature environment of the Liaocheng base, the yields of strain C and D were consistent, at 4620.54 kg/ha and 4603.69 kg/ha, respectively. However, in the harsh environment of Baoshan base, strain D had a higher yield (2664.32 kg/ha), notably higher than strain C (496.67 kg/ha) (Figure 1d, Appendix A). The results of field cultivation also indicated that strain D has stronger thermotolerance.

### 3.2. Common Responses of M. sextelata Strains Under Heat Shock Based on Transcriptome and Proteome Analysis

#### 3.2.1. Transcriptome Analysis

To explore the transcriptional response to heat shock in *M. sextelata*, transcriptome sequencing was performed on the 1st, 3rd, and 5th generation samples of two strains cultured at high temperature (28 °C) and normal temperature (18 °C). After removing adapters and low-quality sequences, a total of 208.02 Gb of clean data were obtained, with Q30 ≥ 91.58%, and the average Q30 at 95.21% ± 0.6 (Appendix A). The average alignment rate of reads to the reference genome is 97.85%, while the expression of 8564 genes is ascertained through the quantification of transcriptomic abundance utilizing FPKM values (FPKM ≥ 10) (Appendix A). The principal component analysis (PCA) result showed that the normal-temperature samples of the two strains significantly clustered together. Under high temperatures, samples of strain C from different periods are separated along Principal Component (PC) 1, while the strain D mainly separated along PC2 (Appendix A), indicating that heat shock has a significant mechanism in both strains, but the involved response patterns may differ. The clustering analysis of the whole-transcriptome data also separated high-temperature samples and normal-temperature samples into two groups, similarly revealing the impact of temperature on the samples (Appendix A).

To explore the basic regulatory patterns of *M. sextelata* in response to high-temperature stress, differential expression analysis was employed to study the transcriptional level differences between high- and normal-temperature samples at each stage, during which the gene expression differences between strains were ignored. In all the three periods of subcultured mycelial samples, there was no significant difference in the number of DEGs that upregulated under heat shock, ranging from 820 to 853. On the other hand, the number of downregulated DEGs (|Log_2_FC| ≥ 1, *p* value ≤ 0.05) exhibited a decrease in the second and third periods, from 778 to around 538 (Figure 2a). The upregulated and downregulated genes were displayed separately through a Venn diagram showing the intersection of DEG sets among the three periods. A total of 222 and 172 genes were commonly upregulated and downregulated across the three periods, respectively. A large number of genes were uniquely highly expressed in a single period, with 381, 260, and 296 genes uniquely upregulated in each of the three periods, indicating that gene expression patterns differ to some extent (Figure 2b). All these upregulated and downregulated genes constitute a non-redundant gene set consisting of 2839 DEGs.

To investigate the gene expression patterns of morel under sustained heat shock, the non-redundant gene set composed of 2839 DEGs was clustered with gene expression levels from N1 samples as the baseline. All genes were divided into four clusters, among which Cluster_1 represents the set of genes downregulated under high temperature, and Cluster_2, Cluster_3, and Cluster_4 corresponded to the sets of genes upregulated during the different periods of high temperature, respectively (Figure 2c). The diversity in the expression patterns of upregulated gene sets imply that *M. sextelata* may face different types of stress pressures at various cultivation stages of heat shock, and it may primarily respond to these different stresses for aiding their survival by enhancing the expression of specific gene sets. Subsequently, these four gene sets were analyzed through GO terms and KEGG clustering. The enrichment results of Cluster_1 which were downregulated under high temperature indicated that many genes were functionally enriched in membrane and transmembrane transport-related functions, and some genes were enriched in catalytic activity as well as glycolysis/gluconeogenesis pathways (Figure 2d). The enrichment results of the high-expression gene sets under high temperature (Cluster_2, Cluster_3, Cluster_4) indicated that the cell activated the Rho protein signaling pathway at the initial stage of high-temperature stress (Cluster_2), while enhancing functions related to ensuring the correct replication and expression of genes, such as RNA splicing and base excision repair (Figure 2e). Subsequently, in the middle period (Cluster_3), the expression of a large number of pathways with functions of maintaining intracellular redox stability increased markedly, such as response to oxidative stress, peroxidase activity, and glutathione metabolism (Figure 2e). In the later stages of stress (Cluster_4), a large number of genes related to ribosome and translation functions were highly expressed, suggesting an increased demand for protein synthesis by the cell at this stage (Figure 2e).

#### 3.2.2. Proteome Analysis

Proteomic sequencing analysis of the 1st, 3rd, and 5th generation samples from two strains cultured under high temperature (28 °C) and normal temperature (18 °C) stress was conducted to explore the impact of high temperature on *M. sextelata* at the protein level. A total of 5453 proteins, with a minimum of 4824 proteins, were identified from six samples (Appendix A). The identified proteins covered 25 out of the 26 functional categories in the COG (Clusters of Orthologous Groups) database, encompassing a wide range of protein types (Appendix A). PCA analysis showed that all samples were initially separated by different temperatures, with all samples cultured at normal temperature significantly clustering together on the right side of PC1, and high-temperature strains on the left side of PC1. Strain D was mainly located in the upper half of PC2, while strain C was mainly located in the lower half of PC2, indicating significant inter-group differences between sample groups of different temperatures and strains (Appendix A). Subsequent whole-proteome clustering analysis also revealed the impact of temperature on the samples, with all samples at normal temperature clustering together and separating from those at high temperature (Appendix A).

To assess the effects of sustained heat shock on the proteome of *M. sextelata*, a comparative analysis was conducted to identify differentially expressed proteins (DEPs) between heat-stressed samples (H1, H2, H3) and their respective controls (N1, N2, N3). The results showed that the number of DEPs upregulated under high temperature was 1144, slightly higher than the number of DEPs downregulated (989) (Figure 3a). The upregulated and downregulated genes were separately displayed through a Venn diagram showing the intersecting relationships between the DEPs of the three periods (Figure 3b). The number of DEPs with consistent expression levels across the three periods was greater than the number of DEPs uniquely expressed in any single period, showing a certain degree of stability across a temporal gradient. However, there were still a large number of proteins that are only upregulated or downregulated in a specific period, especially in the initial and final periods, suggesting that cells have different protein response mechanisms at the beginning and later heat shock. All these upregulated and downregulated DEPs were combined to form a non-redundant DEPs set containing 3414 proteins. The protein expression patterns of these DEPs under continuous heat shock were analyzed by using the Mfuzz tool with protein expression levels from N1 samples as the baseline (Figure 3c). All proteins were divided into four clusters, among which Cluster_1 and Cluster_2 represents the set of proteins downregulated under high temperature, and Cluster_3 and Cluster_4 correspond to the sets of proteins upregulated in the last and first periods under high-temperature stress, respectively (Figure 3c).

Subsequently, GO terms and KEGG enrichment analyses were used to analyze the functions of these four sets of genes. The enrichment results of DEPs in the four clusters indicated that the first period (Cluster_4) of heat shock upregulated the ability of substance transport, involving ammonium transmembrane transporter activity, SNARE interactions in vesicular transport, endocytosis, etc. (Figure 3e). As the heat shock continued, the cell not only made further adjustments to energy metabolism by upregulating pathways such as the tricarboxylic acid (TCA) cycle, carbon metabolism, and pentose phosphate pathway but also enhanced protein synthesis and processing capabilities through upregulating protein processing in endoplasmic reticulum, biosynthesis of amino acids, and protein folding (Figure 3e, Cluster_3). At the same time, the cell maintained redox homeostasis through strengthening the glutathione metabolism, selenocompound metabolism, and cell redox homeostasis pathways (Figure 3e, Cluster_3). The downregulated protein set (Figure 3d, Cluster_1) was functionally enriched in cell membrane-related functions, such as steroid biosynthesis, membrane protein complex, and biosynthesis of unsaturated fatty acids, indicating that the cell membrane may be affected at the whole process of heat shock. The protein set (Figure 3d, Cluster_2), with greatest downregulation observed in the first period (H1), was significantly enriched in pathways related to protein translation and synthesis, such as translation, ribosome, and N-Glycan biosynthesis, indicating that the protein translation process was suppressed throughout the stress process. Among these pathways, the ribosome was extremely significantly enriched (*p* value = 1.62 × 10^−31^), suggesting that the protein synthesis may be inhibited and may be primarily related to the downregulation of ribosome expression.

### 3.3. Comparative Analysis of Thermotolerance Differences Between Strains C and D

To investigate the reasons for the differences in heat tolerance between the two strains, this study first comparatively analyzed the differential expression of the two strains under normal temperature cultivation conditions. The results of DN (all D strain samples cultured at 18 °C) vs. CN (all C strain samples cultured at 18 °C) showed that there were 722 DEGs and 1159 DEPs between the two strains. Functional enrichment analysis was performed on these 722 DEGs and 1159 DEPs. The results of the secondary GO term enrichment indicated that a significantly higher number of contributions were observed in the downregulated genes in the D strain (Appendix A). Moreover, a large number of genes/proteins were enriched in metabolic processes, cellular processes, catalytic activity, and binding (Appendix A). The tertiary GO term results showed that the downregulated genes were mainly enriched in ribosomal and protein synthesis-related pathways. KEGG enrichment analysis also revealed that a portion of the significantly downregulated DEGs were enriched in non-homologous end joining, tyrosine metabolism, and base excision repair (Appendix A). In contrast, the upregulated genes were mainly enriched in transmembrane transport and membrane components (Appendix A).

In order to elucidate the mechanism of the enhanced thermotolerance of strain D, the expression differences in genes and proteins between the two strains under sustained heat shock need to be further explored. By utilizing the Mfuzz clustering algorithm, with the REL values from period N1 as the initial values, time-series clustering of REL across the three periods under heat shock facilitated the identification of RNA sets or protein sets that were upregulated in strain D relative to strain C under high-temperature conditions. Both the RNAs and proteins were categorized into 24 clusters based on the changing patterns of their REL (Appendix A). In time-series clustering analysis of RNA and protein expression profiles, three distinct clusters comprising 574 genes and five clusters encompassing 441 proteins demonstrate a pronounced downward trend, suggesting a reduced expression pattern in strain D when subjected to high-temperature stress. Functional enrichment analysis of the downregulated RNAs and proteins reveals disparities. The downregulated RNAs are predominantly associated with structural constituent of ribosome and translation processes (Appendix A), whereas the proteins are chiefly enriched in ion binding and membrane-related functions, specifically as intrinsic components of the membrane (Appendix A). In total, 14 clusters (seven clusters belong to RNAs and seven belong to proteins) showed an increasing trend in REL for strain D under continuous temperature stress (Figure 4a,b), yielding 1263 genes and 587 proteins, respectively. The GO enrichment and KEGG enrichment analyses of these gene and protein sets indicated a high degree of functional similarity. Significantly enriched GO terms were mainly related to redox processes, such as catalytic activity, oxidoreductase activity, and oxidation-reduction processes (Figure 4c), while significantly enriched KEGG pathways were focused on metabolic pathways, including glycolysis/gluconeogenesis, pyruvate metabolism, and valine, leucine, and isoleucine degradation (Figure 4d). The above results indicated that strain D exhibits superior energy metabolism and redox capacity under heat shock compared to strain C.

Based on differential expression analysis and WGCNA [34], the STRING [35] database, in collaboration with Cytoscape [36], was utilized to identify key elements and functions in gene and protein sets significantly associated with strain D under high temperature. The genes and proteins in strain D that exhibited differential expression between high- and normal-temperature samples in any period were considered to be potentially temperature-related. After filtering out genes and proteins that did not show differential expression under high temperature in any period between the two strains, a total of 1624 genes and 2011 proteins were obtained. Based on 1624 DEGs and 2011 DEPs, with DN serving as the temperature control and CH as the strain control, we categorized the samples into three groups: DH, DN, and CN. Subsequently, we identified the module most positively correlated with DH using WGCNA. The MEturquiose module showed the highest positive correlation with DH in both the gene (*r* = 0.79, *p* value = 1 × 10^−6^) and protein (*r* = 0.83, *p* value = 8 × 10^−8^) (Figure 4e,g). To encompass all protein and transcript information positively correlated with DH, this study integrated the gene sets from the aforementioned two modules to obtain a positively correlated dataset consisting of 1159 genes (Figure 4f). The STRING database was used to construct a protein–protein interaction (PPI) network (Appendix A), and the cytoHubba plugin in Cytoscape (version 3.9.195) with the EcCentricity algorithm was utilized to extract the top 30 genes as hub genes and to map their PPI network. These genes were clustered into three functional groups related to protein regulation: ubiquitin-mediated proteolysis, zymogen, and proteasome regulatory particle binding (Figure 4h). Interestingly, *Rsp5*, a member of the ubiquitin ligase family, demonstrated the highest degree of protein connectivity. Evidence from prior research indicates that it may potentially modulate the expression of heat shock proteins (HSPs) [39], suggesting that *Rsp5* could play a pivotal role in the thermotolerance observed in strain D.

To further verify that *Rsp5* is indeed upregulated exclusively in strain D under heat shock, the expression of *Rsp5* and related HSP genes (*HSP60*, *HSP78*, and *HSP104*) in both strains was further validated using RT-PCR. The results showed that the expression levels of the *Rsp5* gene (Figure 4i) and HSPs (Figure 4j-l) in strain D were significantly higher under high temperature compared to strain C, and the majority of genes showed a significant increase in expression levels compared to normal temperature. These findings provide further validation that the upregulation of *Rsp5* is a distinctive response of strain D to heat shock and may constitute a pivotal factor contributing to its enhanced thermotolerance.

## 4. Discussion

### 4.1. Integration of Transcriptome and Proteome Reveals Key Heat Shock Response Pathways

The mean spearman rank correlation coefficient (*ρ*) for all samples was 0.44 ± 0.05, with the minimum value reaching 0.35 (Figure 5a, Appendix A). This suggested a moderate correlation between the transcriptome and proteome, which could be attributed to the posttranscriptional regulation. To elucidate the discrepancies and interconnections between the transcriptome and proteome, we initially constructed a nine-quadrant plot (Figure 5b). Within this plot, genes exhibiting positive correlations between transcriptional and protein levels were situated in quadrants 3 and 7, whereas those displaying negative correlations were found in quadrants 1 and 9. The genes in these four quadrants exhibited substantial alterations under heat shock. Subsequent functional enrichment analysis, employing KEGG pathway analysis for the genes in these quadrants, indicated that the gene sets in quadrants 1 and 9 were significantly enriched in the ribosome (*p* value = 1.1 × 10^−72^) and glutathione metabolism (*p* value = 2.5 × 10^−3^) pathways, respectively (Figure 5c, Appendix A).

The significant enrichment of ribosome in the first quadrant demonstrates a notable mismatch between transcription and protein levels, which is intuitively displayed in the heatmap constructed based on the log_2_FC values obtained from H vs. N (Figure 5d). This is consistent with the previous results of separate enrichment analyses for significantly upregulated genes (Figure 2e) and significantly downregulated proteins (Figure 3d), both of which were significantly enriched in ribosome and translation pathways. Translation, a process that consumes a large amount of energy, is considered to be one of the earliest processes to be downregulated when the temperature rises [40]. This may be due to that heat shock may trigger 5′-ribosome pausing, leading to a decrease in translation activity [40]. Previous studies have shown that during heat shock, mRNAs encoding ribosomal proteins (RPs) are preferentially stored, and these mRNAs are released and translated during the recovery period [41]. Therefore, it is speculated in this study that a large number of RPs’ mRNAs are stored, leading to a sharp increase in transcription levels. However, since the strains are under continuous heat shock, these mRNAs cannot be released and translated, resulting in a significant decrease in the expression levels of ribosome-related proteins.

Based on previous studies, heat shock proteins (HSPs) and the cell wall integrity (CWI) pathway were considered, as it had been proposed that the heat shock response in fungi is frequently associated with HSPs [42] and can be mediated through the CWI pathway [43]. In *M. sextelata*, 11 HSPs were identified (Appendix A), including *Hsf1* and all family members of HSPs, as well as 11 key genes of the CWI pathway (Appendix A), comprising three upstream sensors and downstream transcription factors, using homologous annotation. Further analysis found a considerable number of significantly upregulated (*p* value ≤ 0.05) HSP genes (six out of 11) and proteins (eight out of 11), as well as CWI key genes (four out of 11) and proteins (six out of 11) under heat shock. The upregulation significance of these two pathways also passed Fisher’s test (*p* value ≤ 0.05), indicating that both play a pivotal role in the comprehensive response of *M. sextelata* to heat shock. In summary, four pathways significantly associated with high-temperature stress were identified: ribosome, glutathione metabolism, HSPs, and CWI. The log_2_FC values of transcripts and proteins for the genes related to these pathways were used to create heatmaps for a visual observation of changes in gene expression levels (Figure 5d–g). The numerous upregulated genes/proteins under high temperatures, which were indicated by red rectangles in the heatmaps, demonstrate that these four pathways are indeed significantly affected by heat shock and may be key pathways for *M. sextelata* adaptation to heat shock.

### 4.2. General Patterns of M. sextelata Response to Heat Shock

Morel is one of the rare cultivable macroascomycetes, its systematic regulatory patterns and networks for high-temperature adaptation remain largely uncharted. In eukaryotes, high-temperature adaptation frequently involves a rapid response mediated by heat shock transcription factors (HSFs) [44], which regulate the expression levels of heat shock proteins (HSPs). HSPs, as molecular chaperones, are involved in protein folding, stabilization, transport, and degradation, thereby assisting in maintaining cellular homeostasis under heat shock [45]. Consistently, our results revealed significant increased expression of the heat shock protein family under high temperatures (Figure 4f) and a higher protein expression of genes related to heat shock protein binding function in strain D compared to strain C under high temperatures (Figure 5c). This suggests that HSPs play a pivotal role in the heat shock response of *M. sextelata* and may contribute to the higher thermotolerance observed in strain D. Additionally, this study observed significant effects of high-temperature stress on the cell wall integrity (CWI) pathway, ribosome, and glutathione metabolism, which corroborates previous findings that these modules can influence HSP expression.

Previous studies have demonstrated that heat shock triggers the CWI-MAPK cascade [46], and the *Rho* gene, which exhibits the highest expression during the initial phase under heat shock (Figure 2e), has been identified as a key component of the CWI pathway [47]. This suggests that the heat shock response in *M. sextelata* may initially activate the CWI pathway. However, the MAPK pathway only begins to exhibit significant high expression in the second phase, possibly due to the partial association of CWI with MAPK pathway genes, which may further contribute to the significant high expression of the MAPK pathway in the second phase for various stresses, such as oxidative stress. The MAPK pathway is known to promote HSP expression [48], and HSPs interact with the CWI pathway to help maintain cell wall stability [49], indicating a positive correlation between the two pathways. The inhibited synthesis of ribosomes (Figure 4d) and the low expression of proteins associated with the cis-Golgi network (Figure 3d) suggests difficulties in protein synthesis under high temperatures. This, in turn, promotes the release of HSFs and subsequent HSPs expression [44,45].

*M. sextelata* significantly upregulated the glutathione metabolism pathway under high temperatures, as evidenced by multiple data analyses (Figure 2e, Figure 3e and Figure 5e). Thirty-two transcripts and twenty-nine proteins associated with glutathione metabolism, comprising a total of fifty-two genes, which include the canonical Glutathione Synthetase (GSS) and Glutathione S-Transferase (GST) enzymes, exhibited upregulation under elevated temperature conditions (Figure 4e). Further exploration of the antioxidant defense system in fungi revealed that, in addition to the glutathione system, the expression of superoxide dismutase [Cu-Zn] (SOD), catalase, and the thioredoxin system (including thioredoxin, thioredoxin reductase, and peroxiredoxin) was also significantly enhanced (Appendix A). The heightened expression of the antioxidant system indicated severe oxidative stress in cells under sustained high-temperature stress, consistent with the transcriptomic observations of *M. sextelata* under heat shock [19]. However, the role of key regulatory factors in the oxygen stress response of *M. sextelata* warrants further investigation. Previous studies have highlighted the significant roles of three main regulatory factors in fungal oxygen stress responses: the HOG gene in the high osmotic glycerol pathway, the yeast activator protein 1-like (*Yap1*) transcription factor containing a basic-leucine zipper (bZIP) domain, and the TF *Skn7* dependent on the osmotic sensing factor [50,51]. In this study, *M. sextelata* lacks the *Atf1* enzyme dependent on the HOG pathway, and *Yap1* is significantly downregulated under heat shock (transcript FC = 0.76, FDR = 7.93 × 10^−7^; Protein FC = 0.63, FDR = 3.45 × 10^−10^), while only *Skn7* is significantly upregulated under heat shock (transcript FC = 1.27, FDR = 0.03; no protein detected), and the transcripts or proteins of the upstream genes *Sln1* and *Ypd1* in the *Skn7* pathway are all significantly highly expressed. These results suggest that the oxygen stress response in fungi may rely on the *Skn7*-related pathway, and previous transcriptional studies have shown that *Skn7* can also activate the expression of heat shock protein-related genes (such as the Hsp70 family member *SSA1*) under oxygen stress [50], further emphasizing the importance of the *Skn7* pathway in the heat response of *M. sextelata*.

Based on these findings, the molecular basis of high-temperature regulation in *M. sextelata* under heat shock has been deduced (Figure 6). High temperatures activate the CWI-MAPK cascade, with the MAPK pathway promoting HSP expression and HSPs interacting with the CWI pathway to maintain cell wall stability. Concurrently, high temperatures inhibit protein synthesis and induce the production of reactive oxygen species, triggering oxidative stress responses. Both protein synthesis inhibition and oxidative stress responses stimulate the expression of heat shock elements. Throughout the entire heat shock response process, HSPs play an indispensable role in *M. sextelata’*s response to heat shock. Western Blot results for strain C further confirm that Hsp60 is indeed significantly highly expressed under high-temperature conditions (Appendix A), providing additional evidence that HSP proteins may play a pivotal role in the high-temperature stress response of *M. sextelata*.

### 4.3. The Strain-Specific Antioxidant and Protein Regulatory Capabilities May Serve as Key Determinants of High-Temperature Tolerance

While the general pattern of high-temperature adaptation in *M. sextelata* has been revealed, the factors contributing to the differential thermotolerance among strains remain a big question. The PCA results showed that the expression patterns of the two had smaller differences at normal temperature. However, the expression differences under normal conditions might, to some extent, reveal the inherent differences due to their origins. The differential expression analysis results under normal conditions indicate that strain D may have a stronger membrane structure, while strain C has vigorous metabolism at the appropriate temperature, which may lead to a higher level of oxidative stress, making it difficult to maintain normal physiological conditions in face of sudden thermal stress.

Compared to the normal temperature culture conditions, the differences in transcription and protein levels between the two strains were more significant under high-temperature stress. The oxidoreductase activity and redox processes in strain D under high temperatures were significantly enhanced at the transcriptional and protein levels (Figure 5c), indicating its superior antioxidant capacity. Compared to strain C, strain D also showed significant enhancements in energy metabolism and amino acid metabolism (Figure 5d). Energy metabolism primarily involved carbon metabolism, glycolysis/gluconeogenesis, and the TCA cycle. Amino acid metabolism mainly involved tyrosine metabolism, biosynthesis of valine, leucine, and isoleucine, as well as phenylalanine metabolism. It is noteworthy that the enhancement of tyrosine metabolism is consistent with previous preliminary observations that tyrosine metabolism may enhance the thermotolerance of Morchella sextelata mycelium [19].

The protein–protein interaction network and DBSCAN clustering of the 30 most relevant genes in strain D under high temperatures revealed a strong association with ubiquitin-mediated proteolysis, zymogen, and proteasome regulatory particle binding, all of which are linked to protein regulatory functions, such as the clearance of damaged proteins, which is a key role of the proteasome [51]. This indicates that protein regulatory capabilities significantly influence the cellular state of strain D under high temperatures. Notably, due to *Rsp5’*s widespread association with other proteins, it may play a significant role in the process. *Rsp5* has been identified as a major E3 ligase that regulates the ubiquitination and proteasomal degradation of heat-induced misfolded proteins [52]. *Rsp5* not only associates with heat-induced misfolded proteins in the cytoplasm [52], facilitating their recognition and degradation, but also targets misfolded proteins at the plasma membrane [53]. Notably, Rsp5 affects rRNA processing and transport [54], thereby contributing to cellular protein homeostasis. The overexpression of *Rsp5* in yeast enhances cellular thermotolerance [55,56]. The Rsp5-Bul1/2 complex has been reported as playing a specific role in *Hsf1*-mediated gene expression and promoting the expression of HSPs [39]. Therefore, the increased expression of the *Rsp5* gene may play a key role in the enhanced thermotolerance observed in strain D, and its impact on HSPs is integrated into the conceptual model of high-temperature regulation in *M. sextelata* (Figure 6).

Based on our current findings, we can only speculate about the underlying mechanisms of differential temperature tolerance between the two strains at transcriptional and proteomic levels. The evolutionary adaptation of fungi to their habitats may serve as one of the driving forces behind their potential temperature tolerance, often involving genetic or epigenetic differences among strains [57,58]. These intrinsic differences may not be significant under normal conditions but can be activated or amplified under stress conditions, leading to specific responses among different strains. Further research is needed to elucidate the molecular mechanisms underlying the differential responses of *M. sextelata* to thermal stress.

## 5. Conclusions

This study charts the first comprehensive conceptual model of heat adaptation mechanisms in *M. sextelata* at the transcriptional and translational levels, providing substantial theoretical and practical insights for the genetic improvement and molecular-assisted breeding of ascomycetes, including *M. sextelata*. The findings indicate that *M. sextelata* activates the CWI-MAPK cascade under high-temperature stress, leading to the generation of reactive oxygen species (ROS) that trigger oxidative stress responses and inhibit protein translation, thereby promoting the synthesis of HSPs. The thermotolerant strain D not only has robust energy and amino acid metabolic capabilities but also shows a stronger antioxidant capacity compared to strain C. It uniquely upregulates genes related to the ubiquitin–proteasome system, including the high expression of the *Rsp5* gene, which may further enhance the expression of heat shock proteins (HSPs) and could be a key factor contributing to its greater thermotolerance. Subsequent RT-PCR experiments confirmed the specific and significant upregulation of *Rsp5* in strain D under heat shock, suggesting its potential as a marker for evaluating *M. sextelata’*s heat adaptation capacity and providing a theoretical basis for selecting thermotolerant strains of *M. sextelata*. In conclusion, by dissecting the complex molecular responses of *M. sextelata* to high temperatures, this study not only provides unprecedented insights into the temperature stress adaptation mechanisms of ascomycetes, as represented by *M. sextelata*, but also provides a crucial scientific foundation for the selection of thermotolerant strains of morel and the development of strategies to improve the stability of morel cultivation.

## Figures and Tables

**Figure 1 jof-11-00076-f001:**
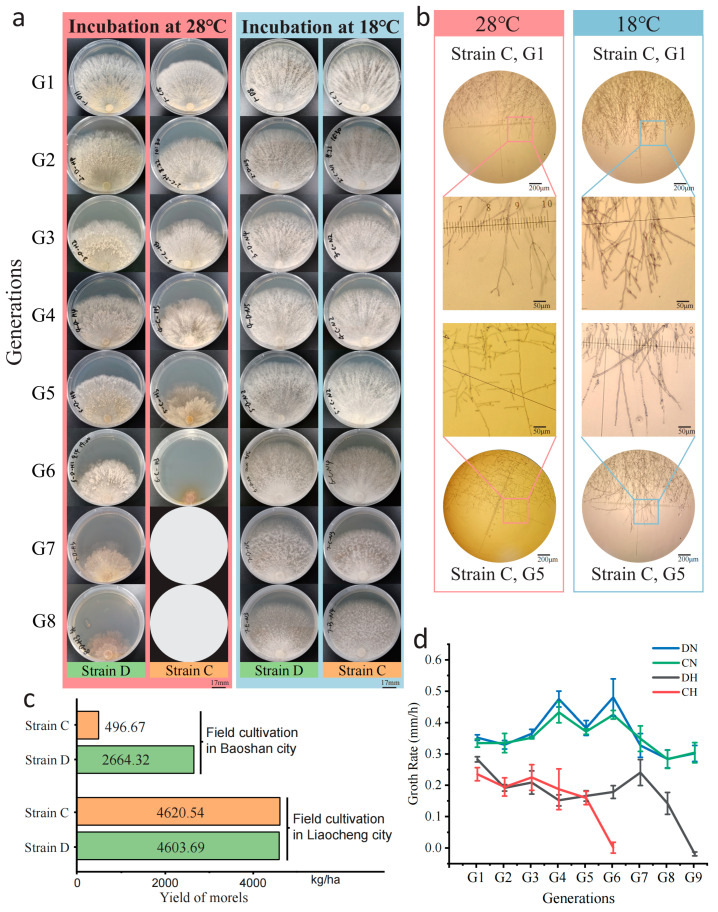
Early mycelial senescence and reduced yield of *M. sextelata* under heat shock. (**a**) The mycelial morphology of *M. sextelata* G1–G8 (strain C) was observed through continuous subculturing at 28 °C (pink) and 18 °C (light blue). Growth of strain C stopped at the sixth generation, and blank circles were used to represent the absence of mycelium in the seventh and eighth generations. (**b**) Microscopic observation of the mycelium of strain C at G1 and G5 under two temperatures (pink, 28 °C; light blue, 18 °C), with selected rectangular frames magnified for display. (**c**) The yield of strains C and D in extreme environment (Baoshan city) and suitable environment (Liaocheng City). (**d**) The growth rate of *M. sextelata* mycelium on PDA medium from the 1st to the 9th generation.

**Figure 2 jof-11-00076-f002:**
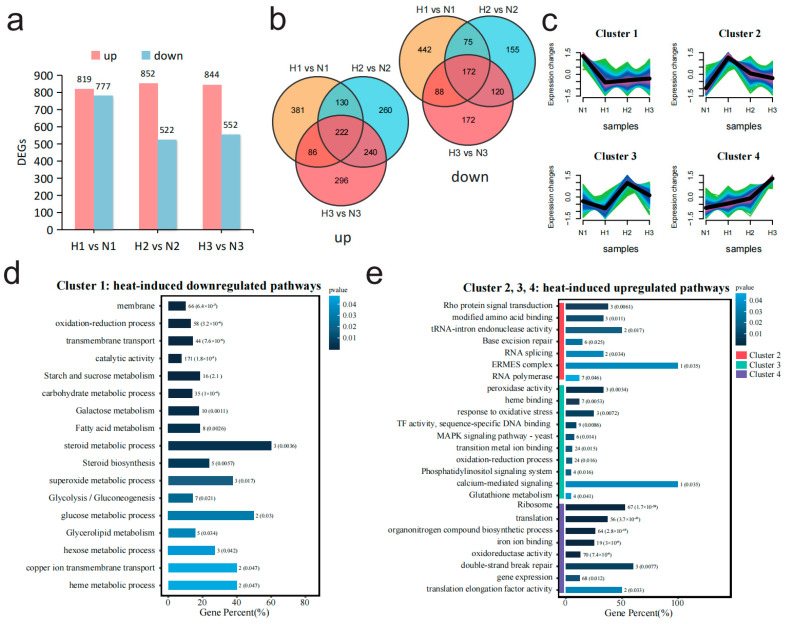
Comparative transcriptomic analysis between high- and normal-temperature conditions reveals the dynamic changes in the transcription levels of *M. sextelata* under high-temperature stress at different time periods. (**a**) The number of up- and downregulated differentially expressed genes (DEGs) in each stage (1, 2, 3) under high- and normal-temperature conditions. (**b**) Two Venn diagrams illustrating the interactive relationships of up/downregulated genes under high-temperature stress. The first Venn diagram displays the intersection of genes that are upregulated (labeled as “up”), while the second Venn diagram shows the intersection of genes that are downregulated (labeled as “down”) during the same periods. (**c**) Mfuzz cluster analysis based on all 2839 non-redundant DEGs in (**b**) divides the genes into four clusters. The color gradient from light to dark (green, turquoise, blue and purple) represents the similarity of gene expression patterns to the cluster center, ranging from low to high. (**d**) Enrichment results of DEGs (Cluster_1) which were downregulated at high temperature. (**e**) Enrichment results of DEGs (Cluster_2, Cluster_3, and Cluster_4) which were upregulated at high temperature.

**Figure 3 jof-11-00076-f003:**
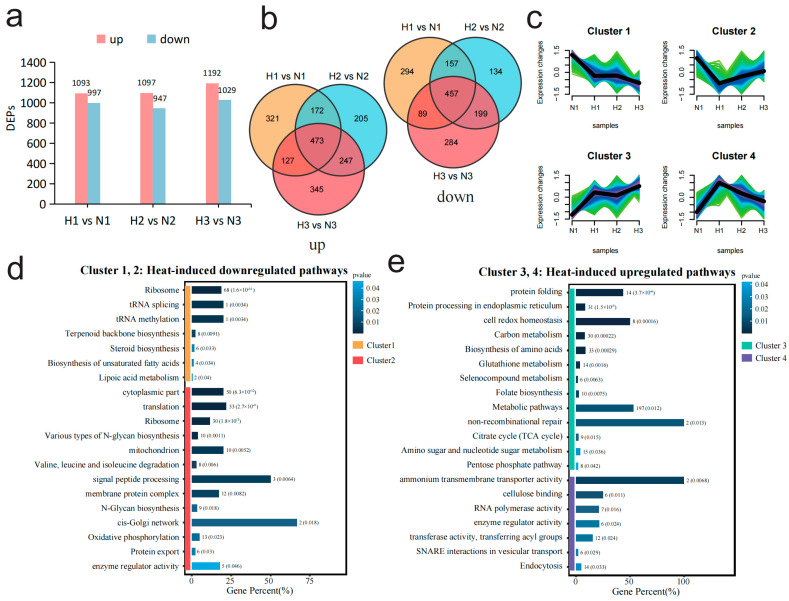
Comparative transcriptomic analysis between high- and normal-temperature conditions reveals the dynamic changes in the proteomic levels of *M. sextelata* under high-temperature stress at different time periods. (**a**) The number of up- and downregulated differentially expressed proteins (DEPs) in each stage (1, 2, 3) under high- and normal-temperature conditions. (**b**) Two Venn diagrams illustrating the interactive relationships of up/downregulated proteins under high-temperature stress. The first Venn diagram displays the intersection of proteins that are upregulated (labeled as “up”), while the second Venn diagram shows the intersection of proteins that are downregulated (labeled as “down”) during the same periods. (**c**) Mfuzz cluster analysis based on all 2839 non-redundant DEPs in (**b**) divides the proteins into four clusters. The color gradient from light to dark (green, turquoise, blue and purple) represents the similarity of gene expression patterns to the cluster center, ranging from low to high. (**d**) Enrichment results of DEPs (Cluster_1 and Cluster_2) with downregulated expression at high temperature. (**e**) Enrichment results of DEPs (Cluster_3 and Cluster_4) with upregulated expression at high temperature.

**Figure 4 jof-11-00076-f004:**
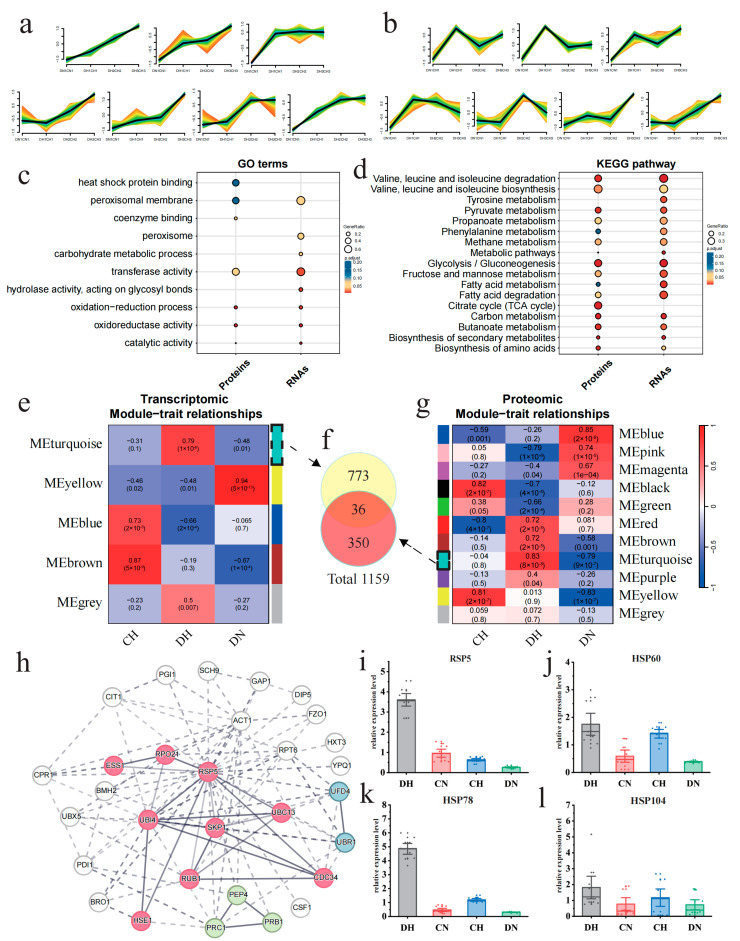
Comparative analysis of thermotolerance differences between strains C and D reveals the differential regulatory pathways and key differentiating factors. (**a**) Clusters exhibiting upregulation under heat shock conditions, derived from trend analysis based on the log_2_FC of transcriptional levels between strains D and C, as shown in the Mfuzz results. (**b**) Clusters exhibiting upregulation under heat shock conditions, derived from trend analysis based on the log_2_FC of protein levels between strains D and C, as shown in the Mfuzz results. In the figures, the color indicates the similarity of genes to the cluster. The color gradient from light to dark (red, yellow, green and turquoise) represents the similarity of gene expression patterns to the cluster center, ranging from low to high. (**c**) GO terms enrichment analysis results for both upregulated proteins and RNAs. (**d**) KEGG enrichment analysis results for both upregulated proteins and RNAs. (**e**,**g**) The WGCNA module–trait correlation heatmaps for the transcriptome and proteome show that the turquoise module has the highest correlation at both the transcriptional and protein levels. In these visualizations, modules are designated by unique colors, with a color bar positioned on one side of the figure and the corresponding color names located on the other side, facilitating clear identification of each module. (**f**) Venn diagram of the intersection relationship of the two most positively correlated modules from WGCNA. (**h**) PPI of the top 30 proteins with the highest connection numbers in the total protein PPI map (Appendix A), which was constructed from 1159 genes coming from (**f**). (**i**–**l**) RT-PCR results of *Rsp5* and three HSP genes.

**Figure 5 jof-11-00076-f005:**
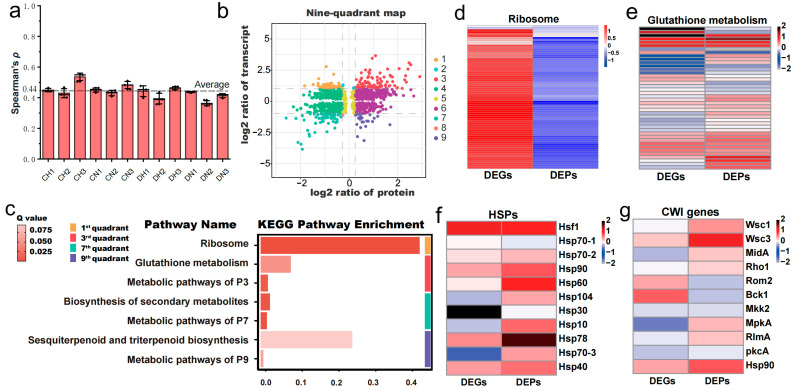
Integrated transcriptomic and proteomic analysis of *M. sextelata* mycelium in response to heat shock. (**a**) Spearman’s rank correlation coefficient between the transcriptome and proteome indicates a moderate correlation (r > 0.4). (**b**) The nine-quadrant plot of the transcriptome and proteome. (**c**) KEGG pathway enrichment results of four important quadrants. (**d**–**g**) Heatmaps generated by the log_2_FC of DEGs and DEPs show the coefficient of variation scale of four critical pathways or gene families under heat shock conditions.

**Figure 6 jof-11-00076-f006:**
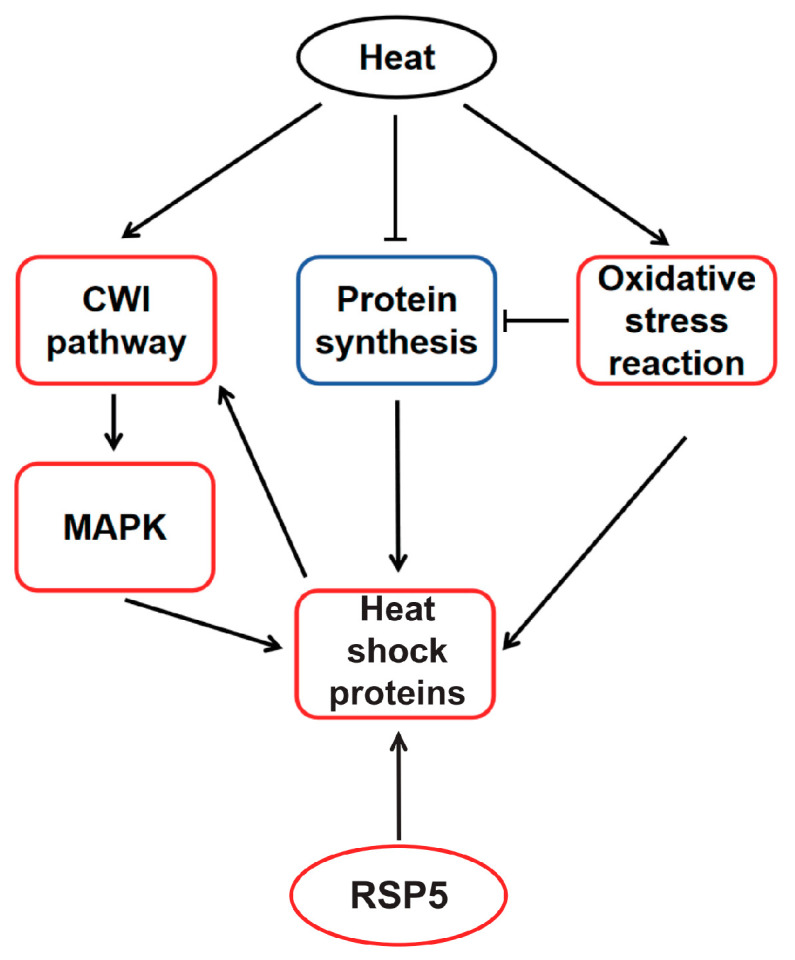
Conceptual model of heat shock adaptation regulatory network. Red box indicates upregulated, blue box indicates downregulated. The activated effects are represented by solid lines with arrows, while inhibitory effects are indicated by arrows with a horizontal bar across them.

## Data Availability

The raw transcriptomic and proteomic data reported in this paper have been deposited in the National Genomics Data Center under the BioProject accession number PRJCA033222. The transcriptomic raw data are available in the Genome Sequence Archive (GSA: CRA021163), which can be publicly accessed at https://ngdc.cncb.ac.cn/gsa (accessed on 12 December 2024). The proteomic raw data are accessible in the OMIX database with the accession number OMIX008225, which can be publicly accessed at https://ngdc.cncb.ac.cn/omix (accessed on 12 December 2024).

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
