# Peer review of "Integrated Transcriptomic and Proteomic Analyses Reveal Molecular Mechanism of Response to Heat Shock in *Morchella sextelata"

_jof, 2025, doi:10.3390/jof11010076_

Round 1
Reviewer 1 Report
Comments:
The article Integrated transcriptomic and proteomic analyses reveal molecular mechanism of heat stress tolerance in Morchella sextelata is very interesting and provides information that helps to better understand the species' biology. He sent some comments:
1. In the methodology section, I suggest putting the information on the isolation process since, as the authors indicate, it is a fungus with complex biology, so it is very important to know how it was isolated since it could affect generations of the strains.
2. The authors comment that they worked with two strains isolated from the natural environment. Still, I did not find the origin of the information since it is very important to have background information.
3. It is essential to mention that M. sextelata is a species adapted to fires; this is a characteristic that indicates its tolerance to temperature stress; however, since there are transcriptomic and proteomic differences between the strains, it could suggest that there are variables of its origin that would be interesting to know and analyze if there is correspondence with the results obtained by the authors.
The writing, in general has no formatting or compression problems.
Author Response
Thank you very much for your insightful and helpful comments. We have revised the manuscript according to your comments and suggestions. Please see the attachment for point-by-point responses.

Reviewer 2 Report
Dear authors! Thank you for the work provided. The article is devoted to the study of molecular mechanisms of morel resistance to hyperthermia. The authors use modern methods of molecular biology and bioinformatics in their work. The relevance of this article leaves no doubt. There are a number of comments. I believe that after revising the manuscript, the article can be published in the journal "JoF". December 27, 2024 Respectfully Yours, reviewer
There are a number of comments: 1. In the Introduction, to enhance the relevance of the research topic, you can write about global warming. There are a huge number of fresh publications on this topic now. 2. I would like to tell the authors that there are 2 concepts: "heat stress" and "heat shock". The authors write about heat stress throughout the paper, however, given the temperature norm for morels and the temperature used in the paper as a heat effect (it is 10 degrees higher than the norm), it is not heat stress, but heat shock. I recommend that the authors understand these terms and choose the right one. 3. Explain in the paper why they used exactly this temperature for stress and exactly this time of heat exposure to create hyperthermia. 4. The Materials and Methods section does not contain a description of the mushroom strains selected for the study (where they were obtained, why these strains were chosen). 5. The Field cultivation validation section is poorly described. According to it, it is unclear how the experiments were carried out, in an open field or in greenhouses. Please clarify. In addition, it is necessary to indicate the characteristics of the soil on which the studies were conducted, the methods of agricultural technology during cultivation and the weather conditions in each of the cultivation seasons. 6. If the authors talk about a change in the content of heat shock proteins in the fungal tissues, it is advisable to confirm this not only on transcript level, but also at the protein level, using the Western Blotting method, which is absent from the paper. 7. Figure 1a is incomprehensible to the reader. 8. The text in almost all the figures is difficult to read. The figures need to be worked on and made more readable. 9. The Results section contains a discussion. I suggest moving the entire discussion to a separate Discussion section, which is in the article. 10. It is necessary to work on the design of the Bibliography according to the MDPI rules; the names of species of living organisms must be written in italics.
Author Response

(The authors gave the same response as above.)

Reviewer 3 Report
The authors in their interesting work using transcriptomic and proteomic analyses performed a comparison between temperature-tolerant and temperature-sensitive strains of Morchella sextelata to identify differentially expressed genes and proteins responsible for heat shock tolerance of M. sextelata in order to provide recommendations for obtaining improved strains.
The authors found an interesting phenomenon of mismatch between transcriptomic and proteomic data. Please add possible explanations of this phenomenon to the discussion. For example, it may indicate the action of posttranscriptional mechanisms of gene regulation, apparently at the level of protein translation. Please look in the literature for examples of genes/proteins that are related to the pathways discovered by the authors, for which the mechanism of mismatch between the change in mRNA and protein levels has been studied in more detail.
Another very interesting observation by the authors is related to the fact that the ubiquitin ligase Rsp5 was found to be involved in the response to heat shock. Please describe in more detail the likely mechanism of its involvement in the regulation of heat shock gene expression in the Discussion section. Please also see if Rsp5 is also targeted in other pathways you have found associated with increased heat shock resistance, such as cell wall integrity pathways?
Line 134. Please specify in more detail how the lysis of M. sextelata cells was performed, i.e. how exactly the cells were disrupted?
Line 142. Please specify the city and country for Bruker.
Line 221. Please specify what G1 means in the phrase “strain D (G1) and strain C (G1)”, maybe the strains should be different in name somehow? Also, G5 is mentioned in the experiments. Please decipher in the text what G1 and G5 stand for.
Author Response

(The authors gave the same response as above.)

Round 2
Reviewer 2 Report
I believe that the article can be published.
I thank the authors for improving the work.